# MFN1 and MFN2 Are Dispensable for Sperm Development and Functions in Mice

**DOI:** 10.3390/ijms222413507

**Published:** 2021-12-16

**Authors:** Junru Miao, Wei Chen, Pengxiang Wang, Xin Zhang, Lei Wang, Shuai Wang, Yuan Wang

**Affiliations:** 1Shanghai Key Laboratory of Regulatory Biology, Institute of Biomedical Sciences and School of Life Sciences, East China Normal University, Shanghai 200241, China; jrmiao30@gmail.com (J.M.); wangpengxiang1900@163.com (P.W.); 13870295778@163.com (X.Z.); wanglei062421@163.com (L.W.); 2Department of Animal Sciences, College of Agriculture and Natural Resources, Michigan State University, East Lansing, MI 48824, USA; chenwe70@msu.edu

**Keywords:** MFN1, MFN2, haploid spermatids, male fertility, sperm functions

## Abstract

MFN1 (Mitofusin 1) and MFN2 (Mitofusin 2) are GTPases essential for mitochondrial fusion. Published studies revealed crucial roles of both Mitofusins during embryonic development. Despite the unique mitochondrial organization in sperm flagella, the biological requirement in sperm development and functions remain undefined. Here, using sperm-specific Cre drivers, we show that either *Mfn1* or *Mfn2* knockout in haploid germ cells does not affect male fertility. The *Mfn1* and *Mfn2* double knockout mice were further analyzed. We found no differences in testis morphology and weight between *Mfn*-deficient mice and their wild-type littermate controls. Spermatogenesis was normal in *Mfn* double knockout mice, in which properly developed TRA98+ germ cells, SYCP3+ spermatocytes, and TNP1+ spermatids/spermatozoa were detected in seminiferous tubules, indicating that sperm formation was not disrupted upon MFN deficiency. Collectively, our findings reveal that both MFN1 and MFN2 are dispensable for sperm development and functions in mice.

## 1. Introduction

Each cell contains several hundred to thousands of mitochondria, and each mitochondrion has 2–10 copies of independent mitochondrial DNAs (mtDNAs) [1,2,3]. To allow coordinated responses of individual mitochondria to various developmental cues, physiological changes, and pathological stress, mitochondria need to share their membrane and exchange their contents through continuous fusion and fission, which are collectively known as mitochondrial dynamics [4,5]. In mammals, mitochondrial fusion is mediated by MFN1 and MFN2, two GTPases at the outer membrane, as well as OPA1 GTPase, an inner-membrane GTPase that regulates mitochondrial inner membrane fusion [6,7,8]. Similarly, the mitochondrial fission is catalyzed by a GTPase called DRP1, along with four adaptors (i.e., MFF, MID49, MID51, and FIS1), which recruit DRP1 to mitochondria [6]. Previous reports demonstrated an essential role of mitochondrial dynamics in mammalian development. Deletion of either *Mfn1* or *Mfn2* causes embryonic lethality due to defects in trophoblast development [9]. When conditional knockouts of *Mfn2* or *Mfn1* and *Mfn2* were used to bypass embryonic placental defects, disrupted cerebellum development and cardiac dysfunction were detected [10].

Recent germ cell studies, including ours, show that disturbed mitochondrial fusion by deletion of either *Mfn1* or *Mfn2* specifically blocks spermatogonial differentiation, but undifferentiated spermatogonia remain largely unaffected [11,12,13,14]. The difference in sensitivity to disrupted mitochondrial dynamics reflects the importance of differential mitochondrial regulation in various developmental stages of spermatogenesis. Although the physiological importance of *Mfn1* and *Mfn2* currently remains undefined during late spermatogenesis, several reports suggest that mitochondrial dynamics may participate in sperm development. For example, Honda et al. investigated the *Mfn1* and *Mfn2* expression patterns in rats by Northern blot analysis and found that *Mfn1* and *Mfn2* were highly expressed in the pubertal and adult rat testes. Immunohistochemistry analysis revealed a pronounced expression of MFN2 in the round and elongating spermatids in the seminiferous tubules [15]. In sperm, mitochondria line the microtubular axoneme and wrap around the midpiece of sperm flagellum to form the mitochondrial sheath [16]. MFN2 was also found to be localized at the flagellum midpiece and post-acrosomal regions of sperm in the caput and cauda area of the epididymis by immunofluorescence analysis [16]. In addition, deficiency of mitochondrial fission caused by *Fis1* or *Mff* null mutations leads to defective mitochondria, aberrant sperm function, and reduced fertility [17,18,19]. These data suggest a potential role of mitochondrial dynamics at the post-meiosis stage in male mice.

To understand whether mitochondrial fusion participates in late spermatogenesis, we generated sperm-specific knockout mice for *Mfn1* and *Mfn2* with *Prm1-Cre* drivers. We found that neither *Mfn1* nor *Mfn2* deficiency in haploid sperm caused defects in sperm development and male fertility in mice. Furthermore, to exclude functional redundancy of MFN1 and MFN2, compound deletion of *Mfn1* and *Mnf2* was analyzed, and male fertility remained unaffected, indicating that both genes are dispensable during haploid sperm formation and functions. Our study thus suggests that mitochondrial fusion plays a nonessential role in maintaining sperm development in mice.

## 2. Results

### 2.1. Mfn1 and Mfn2 mRNA Are Highly Expressed in Testes

To understand whether mitochondrial fusion participates in late spermatogenesis, we first examined if *Mfn1* and *Mfn2* were expressed in testes by qPCR analyses on various mouse tissues. We found preferential expression of *Mfn1* in spleen and testes, while *Mfn2* was highly expressed in mouse testes (Figure 1A,C). Because in spermatogenesis, spermatogonia start to differentiate at postnatal day (P) 6–10, with meiotic pachytene spermatocytes emerging at P14, and haploid spermatids around P21, we further analyzed the expression of *Mfn1* and *Mfn2* in mouse testes at different ages by qPCR. We found that *Mfn1* mRNA level started to increase at week 2 and peaked at week 5 (Figure 1B). Similarly, the highest expression of *Mfn2* mRNA level was detected at weeks 4–5 (Figure 1D). These results suggest that *Mfn1* and *Mfn2* were highly expressed in haploid spermatids, corresponding with the time of post-meiotic germ cell development.

### 2.2. Generation of Mfn1 and Mfn2 Conditional Knockout (cKO) Mice

We previously revealed that either *Mfn1* or *Mfn2* conditional knockout in early germ cells resulted in male infertility [11,13]. To understand whether *Mfn1* and *Mfn2* regulate post-meiotic germ cell development, we crossed *Mfn1* and *Mfn2* conditional knockout mice with a *Prm1*-*Cre* line (Figure 2A,B), in which CRE recombinase under the sperm-specific promoter of *Prm1* deletes *Mfn1* or *Mfn2* in haploid spermatids [20,21]. Because antibodies against MFN1 or MFN2 for immunohistofluorescence did not work well in our hands, we examined the *Mfn1* and *Mfn2* deletion efficiency in sperm by the germline transmission of mutant alleles. In this assay, *Mfn1^f/f^*, *Prm1*-*Cre* and *Mfn2 ^f/f^*, and *Prm1*-*Cre* male mice were crossed with wild-type females. If the CRE deletion is incomplete in sperm, some progenies generated from these mice would contain the *Mfn1^f/+^* (or *Mfn2^f/+^*) allele instead of *Mfn1^−/+^*(or *Mfn2^−/+^*). The results demonstrated that 100% of progeny mice generated from paternal *Mfn1^f/f^*; *Prm1*-*Cre* (or *Mfn2^f/f^*; *Prm1*-*Cre*) were carrying the *Mfn1^−/+^*(or *Mfn2^−/+^*) allele. Thus, these results confirm that *Prm1*-*Cre* was effective, deleting floxed *Mfn1* or *Mfn2* alleles with 100% efficiency (Table 1).

### 2.3. Either Mfn1 or Mfn2 Deletion in Sperm Does Not Affect Sperm Development and Functions

Conditional knockouts of *Mfns* offer powerful models to understand the functional contribution of MFNs to mouse sperm development and male fertility. We thus analyzed the sperm development upon deletion of *Mfns* by *Prm1-Cre*.

We found that the testes in *Mfns*-cKO (i.e., *Mfn1^f/f^*, *Prm1*-*Cre* or *Mfn2^f/f^*, *Prm1*-*Cre*) mice were morphologically normal at 56 dpp compared to wild littermate controls (Figure 3A,C). The weight of *Mfns*-cKO testes showed no significant difference from controls either (Figure 3B,D). We next examined germ cell development in control and *Mfns*-cKO mice at 42 dpp by immunohistofluorescence (IHF) with antibodies against TRA98, a pan-germ cell marker [22], and SYCP3, a spermatocyte-specific marker [23]. TRA98 staining revealed normal meiosis and post-meiotic round spermatid development in *Mfns*-cKO mice (Appendix A). Similarly, SYCP3 signals were comparable between *Mfns*-cKO and control mice (Appendix A), indicating a normal spermatocyte development. TNP1, a protein that only marks haploid germ cells, was also readily detectable in testis sections from *Mfns*-cKO mice, similar to those in control littermates (Appendix A), indicating the existence of spermatids and spermatozoa. We further analyzed the epididymides and found no obvious difference in the numbers of spermatozoa in *Mfns*-cKO mice compared to their littermate controls (Figure 3E). Finally, we evaluated the impact of *Mfns* deletion on male fertility using a breeding test, in which *Mfns*-cKO mice and their littermate control male were crossed with wild-type female mice, and the average number of pups per litter over 2 months was calculated. We did not detect any obvious change in numbers of pups per litter produced from breeding pairs with *Mfns*-cKO male mice compared to those from their littermate controls (Figure 3F). Taken together, these results suggest that *Mfns* conditional knockout in post-meiotic haploid germ cells does not affect sperm development and functions in mice.

### 2.4. Combined Deletion of Mfn1 and Mfn2 in Sperm Does Not Affect Sperm Development and Functions

MFN1 and MFN2 share more than 81% homology and similar topology [9,24,25,26,27]. In fibroblasts, MFN1 and MFN2 compensate the function deficiency of mitochondrial fusion for each other [9]. We showed that deletion of either *Mfn1* or *Mfn2* does not affect sperm development and functions in late spermatogenesis. To exclude the possibility that this is due to functional redundancy between MFN1 and MFN2 in spermatozoa, we further generated *Mfn1* and *Mfn2* double knockout mice (*Mfn1^f/f^*, *Mfn2^f/f^*, *Prm1-Cre*, abbreviated as *Mfn1&2*-cKO). Testes from *Mfn1&2*-cKO and control littermates at 56 dpp were analyzed. We did not find any significant differences in morphology and weight of testes between the two groups (Figure 4A,B). Moreover, IHF staining with TRA98, SYCP3, and TNP1 antibodies revealed the presence of all types of germ cells from spermatogonia to spermatozoa in *Mfn1&2*-cKO mice (Appendix A). In addition, mature spermatozoa were readily detected in epididymides of adult *Mfn1&2*-cKO mice (Figure 4C). We further found that deletion of both *Mfn1* and *Mfn2* in haploid germ cells did not affect sperm functions and male fertility. In the breeding assay, the average numbers of pups per litter from adult *Mfn1&2*-cKO mice showed no significant difference from those in the control group (Figure 4D). Taken together, these data demonstrated that the compound knockout of *Mfn1* and *Mnf2* in haploid germ cells did not affect sperm development and male fertility in mice.

## 3. Discussion

Both mitofusins, MFN1 and MFN2, are broadly expressed across various tissues [26]. Recent evidence suggests that MFN1 and MFN2 have both redundant and distinct functions, depending upon cell types and developmental stages [9,28,29,30]. Although roles of mitochondrial fusion in early spermatogenesis before meiosis have been reported, the exact expression pattern and physiological importance of *Mfn1* and *Mnf2* in mouse haploid spermatids have yet to be characterized. Here we examined the functions of *Mfn1* and *Mfn2* in sperm development and male fertility.

Published studies demonstrate that *Mfn1*-null or *Mfn2*-null mouse embryonic fibroblasts (MEFs) contain highly fragmented mitochondria, and MFN1 and MFN2 compensate the function deficiency of mitochondrial fusion for each other [9]. To understand the functional roles of mitochondrial fusion in sperm formation, we generated conditional knockout of *Mfns*, using two haploid-specific *Prm1-Cre* drivers. The first *Prm1-Cre* driver came from Jackson Laboratory and displayed a partial excision of *Mfns* based on our analyses on transmission efficiency of mutant alleles (Appendix A). To examine the effects of complete MFN deficiency on sperm formation, we further used the second *Prm1-Cre* (RIKEN BRC) strain, which efficiently removed *Mfns* from haploid spermatids (Table 1). Notably, although both *Prm1-Cre* mice were generated through the pronuclear microinjection of transgene into fertilized eggs, the regulatory elements of these two *Prm1-Cre* transgenes are different [20,21,31]. In addition, the expression of randomly inserted transgenes at different genomic locations is often subject to chromatin positional effects. The genomic location of both the *Prm1-Cre* (Jackson) and the *Prm1-Cre* (RIKEN BRC) transgene are unknown. It is thus possible that *Prm1-Cre* from Jackson Laboratory is not transcriptionally active in a subset of haploid spermatids. Consistent with our findings on those *Prm1-Cre* mouse models, Batista et al. also observed the inefficient recombinase activity when using *Prm1-Cre* from Jackson Laboratory [32]. However, another study used the same *Prm1-Cre* strain from Jackson Laboratory but did not find such incomplete Cre recombinase activity [33]. One potential explanation for this inconsistent observation may be due to different genomic loci of various floxed alleles [32]. The chromatin status of certain floxed genomic loci may prevent CRE from accessing the loxP sites. Alternatively, a genetic drift of the Cre transgene may occur over time. As a mouse line is continuously intra-bred within the colony, mutations may arise and lead to reduced CRE expression. Therefore, properly maintained and characterized mouse models are crucial to permit successful gene manipulation in reproductive research.

Using these haploid-specific *Prm1-Cre* drivers, our study showed that mice with single deletion of either *Mfn1* or *Mfn2* in spermatids had normal sperm development and male fertility, either when *Mfn*s were partially deleted (Appendix A) or when efficient *Prm1-Cre* driver from RIKEN BRC completely removed *Mfn*s (Table 1, Figure 2 and Figure 3) in sperm. To exclude the possibility of functional compensation between MFN1 and MFN2 in sperm, we further analyzed combined knockouts of *Mfn1* and *Mfn2*, using the efficient *Prm1-Cre* from RIKEN BRC. Unexpectedly, adult mice with complete deficiency of both MFNs displayed well-developed and functional sperm. The male fertility remained unaffected. This is unlikely due to any existing truncated functional MFN proteins in sperm. Deletion of exon 4 of *Mfn1* and of exon 6 in *Mfn2* by CRE recombinase leads to a frameshift of *Mfn* genes. In addition, CRE deletes the exon four or exon six encoding the canonical GTPase motif of MFNs, thus preventing the formation of any functional protein. Notably, the same mouse *Mfn* cKO lines (*Mfn1*: MMRRC # 029901-UCD and *Mfn2*: MMRRC # 029902-UCD) have been widely validated to be absent of functional proteins upon CRE deletion in cells from these mice, including neurons [10], skeletal muscle [34], and early-stage germ cells in our hands [11,13]. Therefore, our data suggest that mitochondrial fusion is dispensable for sperm development during late spermatogenesis.

Mitochondria are dynamic organelles that continually undergo fusion and fission. These two opposing processes work in concert to maintain the shape, size, and number of mitochondria and their physiological functions [6]. Previous evidence suggests that two mitofusins form three distinct molecular complexes to promote mitochondrial fusion: MFN1 homotypic oligomers, MFN2 homotypic oligomers, and MFN1-MFN2 heterotypic oligomers [9]. The type of complex that plays a dominant action depends on the cell type. For example, MEFs appear to use all three oligomeric complexes [9]. By contrast, trophoblast giant cells rely on MFN2 homotypic oligomers. Placental defects were detected in *Mfn2* but not *Mfn1* mutants [9]. Our results suggest none of these complexes are required for proper sperm functions. The morphology, number, and mitochondrial organization of germ cell mitochondria change throughout spermatogenesis. Spermatogonia and early spermatocytes harbor “orthodox” mitochondria, while late spermatocytes and early spermatids contain a condensed form of mitochondria with enlarged matrix and vesiculate cristae, randomly distributing throughout the cytoplasm [35]. In elongated spermatids, mitochondria with “intermediate” configuration are aligned along the outer dense fibers in the forming tail [36,37,38]. During spermiogenesis, a large fraction of mitochondria is degraded, leaving far fewer mitochondria (~50–75 per cell) and mtDNAs (~1.4 per mitochondrion) in the sperm [37,38,39,40]. Although sperm need mitochondrial metabolism for energy production to support their mobility and fertilization, the activities of mitochondrial dynamics in sperm are likely kept minimal to ensure proper mitochondrial organization. Indeed, low levels of mitochondrial fusion and fission in sperm have been reported [41]. Therefore, the haploid spermatids likely have high tolerance for perturbations in mitochondrial dynamics. This is different from differentiating spermatogonial and spermatocytes; loss of *Mfn1* or *Mfn2* alone will result in germ cell development defects, leading to male sterility in mice [11,12,13,14].

## 4. Materials and Methods

### 4.1. Experimental Design

Conditional *Mfn* knockouts by *Prm1-Cre* were obtained and analyzed as following (Figure 5):

### 4.2. Mouse Lines, Animal Care, and Fertility Test

Sperm harboring the floxed *Mfn1* and *Mfn2* alleles were obtained from Mutant Mouse Regional Resource Center (MMRRC at UC Davis, USA, resource identifier no. MMRRC # 029901-UCD and MMRRC # 029902-UCD) [10]. *Mfn1^f/+^* and *Mfn2^f/+^* mice were derived by intracytoplasmic injection of sperm into oocytes and maintained by crossing with C57BL/6 mice. Two *Prm1-Cre* mouse lines were used. One was obtained from Shanghai Model Organisms Center Inc (originally developed from Jackson Laboratory, Bar Harbor, ME, USA, Stock ID: 003328), and another was purchased from RIKEN BioResource Research Center (ID: RBRC02182) [20,21]. To generate the post-meiotic haploid spermatids specific deletion of *Mfn1*, male conditional *Mfn1* knockout mice (*Mfn1^f/+^*) were bred with female *Mfn1^f/+^* and *Prm1-Cre*, and the progeny mice were genotyped with PCR analyses. Conditional *Mfn2* knockout mice were generated in the same way as described above. For the fertility test, adult males were housed with wild-type C57BL/6 females for at least 2 months. The size of the litters generated by these males was recorded. All animal experimental procedures (Protocol ID: m20190403) were conducted in accordance with the local Animal Welfare Act and Public Health Service Policy (consistent with the WMA Statement on Animal Use in Biomedical Research) and approved on 01-04-2019 by the Committee of Animal Experimental Ethics at East China Normal University.

### 4.3. Genomic DNA Extraction and Genotyping

Genomic DNA was isolated as described previously [42,43]. Briefly, tail tips were digested in alkaline lysis reagent (25 mM NaOH, 0.2 mM EDTA at pH = 12) at 95 °C for 30 min, and the reactions were then stopped by neutralizing reagent (40 mM Tris-HCl, pH = 5). The supernatant was collected for PCR genotyping with Taq Master Mix (Vazyme Biotech, Nanjing, China, P112-AA) according to the manufacturer′s instructions. Sequences of primers and size of products are provided in Appendix A.

### 4.4. Total RNA Extraction, Reverse Transcription, and Real-Time PCR

As previously described [42], total RNA from various mouse tissues was extracted using RNAiso Plus solution (TAKARA, Dalian, China, 9109), and cDNAs were synthesized using a cDNA Synthesis Kit (TAKARA, RR037A). Quantitative Real-Time PCR was performed on Thermo Scientific QuantStudio 3 Real-Time PCR System. Three technical replicates were conducted for each independent experiment. Sequences of primers used in this study are listed in Appendix A.

### 4.5. Histology and Immunohistofluorensce (IHF)

IHF was performed as described previously [42]. Testes were fixed in 4% paraforaldehyde (PFA) in PBS at 4 °C overnight and embedded in paraffin. The 5 μm thick testicular sections were incubated with primary antibodies and washed three times with PBS containing 0.1% Tween 20. Nuclei were stained in 0.5 mg/mL DAPI (Beyotime Biotechnology, Shanghai, China, C1002) after blotting with Alexa Fluor^®^ conjugated secondary antibodies. Slides were imaged under a fluorescent microscope (Olympus BX53, Olympus Life Science, Japan) and processed with Image J software. Antibodies and fluorescent probes used in this study: TNP1 (ab73135), TRA98 (ab82527; 1:500), and SYCP3 (ab15093; 1:400) were purchased from Abcam (Cambridge, MA, USA), Alexa Fluor^®^ conjugated secondary antibodies came from Jackson ImmunoResearch Laboratories (West Grove, PA, USA).

### 4.6. Statistics

The data were evaluated for significant differences using Student’s *t*-test, calculated with GraphPadPrism5 software (GraphPad Software, La Jolla, CA, USA). A *p*-value < 0.05 was considered statistically significant.

## Figures and Tables

**Figure 1 ijms-22-13507-f001:**
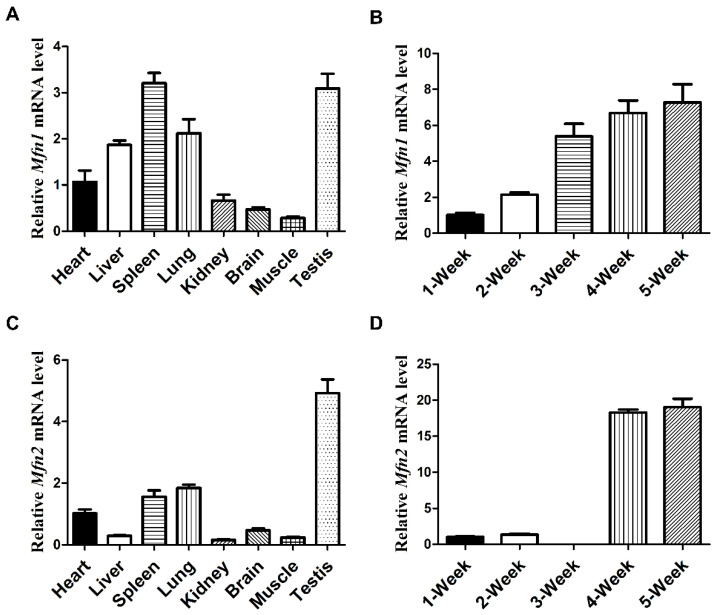
*Mfn1* and *Mfn2* are highly expressed in testes. (**A**,**C**) qPCR analyses of *Mfn1* mRNA (**A**) and *Mfn2* mRNA (**C**) expression levels in various organs collected from wild-type adult mice. (**B**,**D**) qPCR analyses of *Mfn1* (**B**) or *Mfn2* (**D**) mRNA expression levels in developing testes at 1, 2, 3, 4, and 5-week. *Mfn1* or *Mfn2* mRNA expression levels in heart (**A**,**C**) and week 1 (**B**,**D**) were used as baseline control. *Gapdh* served as the cDNA loading control. Data are presented as mean ± SEM.

**Figure 2 ijms-22-13507-f002:**
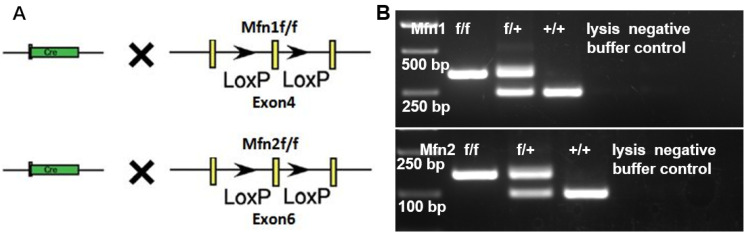
Deletion of *Mfn1* and *Mfn2* in haploid sperm by *Prm1-Cre*. (**A**) Strategy to delete *Mfn1* and *Mfn2* in haploid sperm by *Prm1*-*Cre*. The exon 4 of *Mfn1* and exon 6 of *Mfn2* were flanked by two loxP sequences for deletion by Cre/loxP-mediated recombination. Yellow rectangles represent exons; black arrows represent loxP sequences. (**B**) Examples of genotyping on offspring from *Mfn1^f/+^* and *Mfn2^f/+^* male mice crossed with *Mfn1^f/+^*, *Prm1*-*Cre* and *Mfn2^f/+^*, *Prm1*-*Cre* female. PCR on lysed mouse tail snips amplified the *Mfn1* genomic locus to generate a 450 bp DNA band from the mutated region, or a 350 bp band from wild-type locus. In *Mfn2* mice, PCR amplified a 180 bp DNA band from the mutated region, or a 145 bp band from wild-type locus. +/+: wild-type; f/+: heterozygous; f/f: homozygous.

**Figure 3 ijms-22-13507-f003:**
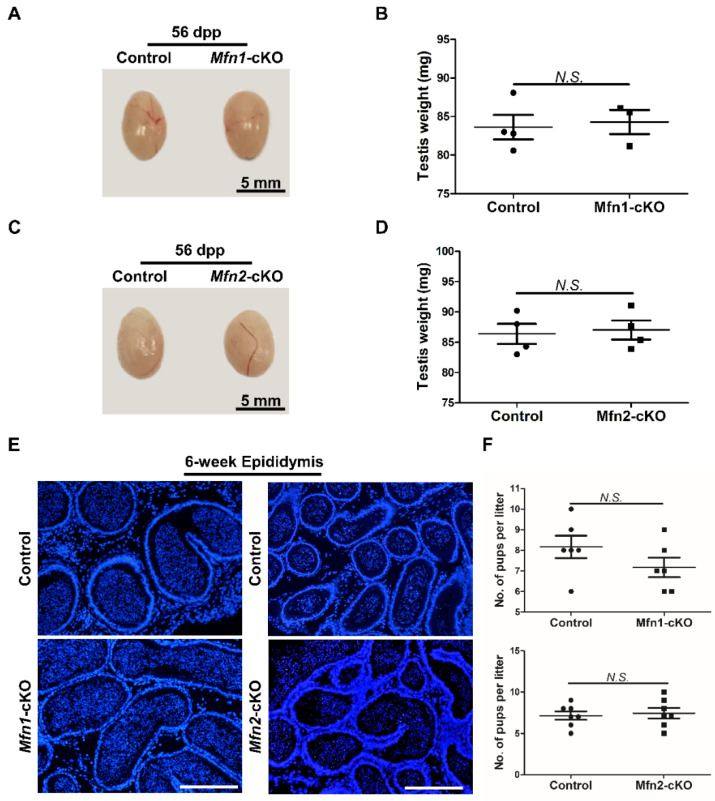
Either *Mfn1* or *Mfn2* deletion in haploid germ cells does not affect sperm development and functions. (**A**,**C**) Gross morphology of testes from control and *Mfn1*-cKO mice (**A**) or *Mfn2*-cKO mice and their littermate controls (**C**) at 56 dpp. (**B**,**D**) Average testis weight was calculated from *Mfn1*-cKO (**B**) or *Mfn2*-cKO (**D**) mice and control littermate at 56 dpp. (**E**) DAPI stained sections of epididymides from control, *Mfn1*-cKO, and *Mfn2*-cKO mice at 6 weeks. Scale bars: 200 μm. (**F**) Average numbers of pups per litter were calculated based on 6 litters from 3 control and 6 litters from 3 *Mfn1*-cKO male mice (upper panel), or based on 7 litters from 3 control and 7 litters from 3 *Mfn2*-cKO male mice (lower panel). Each male mouse was bred with 2 wild-type females for over 2 months. Data are presented as mean ± SEM. N.S.: no significance.

**Figure 4 ijms-22-13507-f004:**
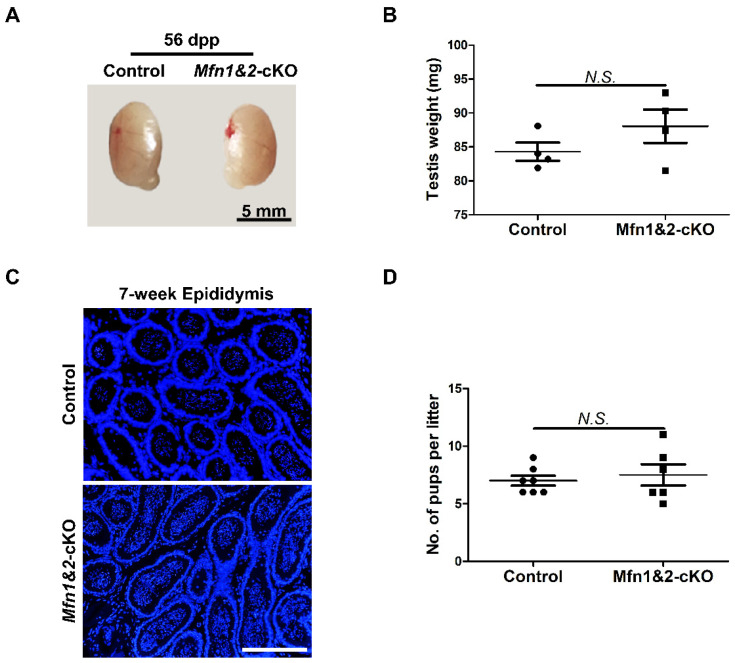
Compound knockout of both *Mfn1* and *Mfn2* in haploid sperm does not affect sperm development and mice fertility. (**A**) Gross morphology of testes from control and *Mfn1&2*-cKO mice at 56 dpp. (**B**) Averaged testis weight was calculated from 4 control and 4 *Mfn1&2*-cKO littermates at 56 dpp. (**C**) DAPI stained sections of epididymides from control and *Mfn1&2*-cKO mice at 7 weeks. Scale bars, 200 μm. (**D**) Averaged pup numbers per litter were calculated based on 7 litters from 3 control and 6 litters from 3 *Mfn1&2*-cKO male mice. Each male mouse was bred with 2 wild-type females for at least 2 months. (**B**,**D**) Data are presented as mean ± SEM. N.S.: no significance.

**Figure 5 ijms-22-13507-f005:**
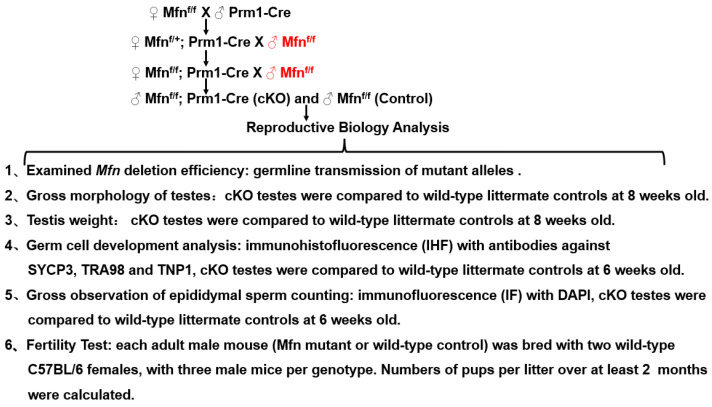
Summary of experimental design and methods used in this study.

**Table 1 ijms-22-13507-t001:** Transmission efficiency of mutant alleles.

Floxed Alleles	NO. of Males	NO. of Litters	NO. of Pups	F/+ Pups	Del/+ Pups	Transmission Del/+ Allele
*Mfn1^f/f^*; *Prm1-Cre*	3	6	43	0	43	100%
*Mfn2^f/f^*; *Prm1-Cre*	3	7	52	0	52	100%

## Data Availability

The data presented in this study are available in the article and Appendix A.

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
