# Peer review of "MFN1 and MFN2 Are Dispensable for Sperm Development and Functions in Mice"

_ijms, 2021, doi:10.3390/ijms222413507_

Round 1

Reviewer 1 Report

“MFN1 and MFN2 Are Dispensable for Sperm Development and Functions in Mice” by Junru Miao and colleagues reports the male postmeiotic germ cell-specific knockout of MFN1 and MFN2 as separate (single) knockouts and combined (double) knockout. MFNs are proteins involved in mitochondrial dynamics and show strong expression in the adult mouse testis suggesting a functional role of these two proteins in spermatozoa, which rely on functional mitochondria. So, the rational of the study is clear and evident.

The study is clear, generally well-designed and the presentation of the data is generally convincing. Also the manuscript is well-written and focusses on the essential points. Unfortunately, no phenotype, neither in the individual knockouts nor in the combined double-knockout was found. Nevertheless, this “negative result” does not influence the scientific value of the study as it falsifies an interesting and reasonable hypothesis.

However, there are a few things that require attention and revision:

  1. It needs to be shown (or at least explained) that no functional protein is left after deletion of the exons 4 and 6, respectively. The genes consist of several exons and it is not clear from the information provided that no functional protein is left. The absence of the protein needs to be shown or an explanation must be provided why no functional protein can be synthesized assuming that the deletion of the gene in postmeiotic cells is complete (which is likely based on the mating data shown in Table 1).
  2. Is anything known about the MFN1 and MFN2 mRNA content in sperm? Can it be excluded that the transcripts play a role during post-fertilization processes? Does the oocyte also contain MFN1 and MFN2 transcripts?
  3. Some words are apparently missing, e.g. line 19, line 44.
  4. Line 64: plays
  5. Line 73: spermatids
  6. Line 77: It is suggested to replace “revealed” with “suggest”.
  7. Lettering of the organ names in Fig.1c is smaller than in the other parts of the figure.
  8. 2b: Please provide the size of the marker bands.
  9. Line 118 ff: TRA-98 and SYCP are nice markers of germ cells and meiotic germ cells, respectively. However, there is no postmeiotic marker shown which would be much more meaningful in the context of this study. Therefore, it is strongly suggested to stain the testes sections with a (late) spermatid marker.
  10. Line 121: We…
  11. Line 169: … meiosis have been…
  12. Line 171: I would recommend not to write that a “detailed” analysis has been performed. The analysis is sufficient to say that the proteins are not important for sperm development and for successful fertilization. However, a detailed analysis would require an in-depth-analysis.
  13. Line 196: …and lead to…
  14. Line 214: “depend”?
  15. Line 216: “om”??
  16. Page 291: As far as this reviewer knows the Declaration of Helsinki applies only to research with human subjects. Therefore, the Helsinki Declaration may not apply to this mouse study.

In summary, this is an interesting data set. However, the manuscript requires a revision.

Author Response

Response to Comments of Reviewer 1

Comments

“MFN1 and MFN2 Are Dispensable for Sperm Development and Functions in Mice” by Junru Miao and colleagues reports the male postmeiotic germ cell-specific knockout of MFN1 and MFN2 as separate (single) knockouts and combined (double) knockout. MFNs are proteins involved in mitochondrial dynamics and show strong expression in the adult mouse testis suggesting a functional role of these two proteins in spermatozoa, which rely on functional mitochondria. So, the rational of the study is clear and evident.

The study is clear, generally well-designed and the presentation of the data is generally convincing. Also, the manuscript is well-written and focusses on the essential points. Unfortunately, no phenotype, neither in the individual knockouts nor in the combined double-knockout was found. Nevertheless, this “negative result” does not influence the scientific value of the study as it falsifies an interesting and reasonable hypothesis.

However, there are a few things that require attention and revision:

Response: Thank you for your nice comments. Please see our responses below to your specific critiques.

Minor comments

Critique 1: It needs to be shown (or at least explained) that no functional protein is left after deletion of the exons 4 and 6, respectively. The genes consist of several exons and it is not clear from the information provided that no functional protein is left. The absence of the protein needs to be shown or an explanation must be provided why no functional protein can be synthesized assuming that the deletion of the gene in post meiotic cells is complete (which is likely based on the mating data shown in Table 1).

Response: We agree with the reviewer that it would be informative to show the absence of MFN proteins in Mfn knockout sperm by immunohistofluorescence (IHF). We previously tried to find commercial MFN1 antibodies suitable for IHF at major antibody suppliers including Abcam (ab57602, ab104274 and ab107129), Santa Cruz Biotechnology (sc-166644, sc-50330), and Abnova (H00055669-M04, MAB10282), but unfortunately none worked for IHF. 

Below a diagram from reference (PMID: 17693261) shows that deletion of exon 4-6 by CRE recombinase leads to a frame-shift of the Mfn genes. In addition, CRE deletes the exon 4 or exon 6 encoding the canonical GTPase motif of MFNs, thus preventing formation of any functional protein. Furthermore, the same mouse Mfn cKO lines (Mfn1: MMRRC # 029901-UCD and Mfn2: MMRRC # 029902-UCD), have been widely validated to be absent of functional proteins upon CRE deletion in cells from these mice, including neurons (PMID: 17693261), Skeletal Muscle (PMID: 20403324), and early-stage germ cells in our hands (PMID: 26711429, and 32330448). We included these explanations in the revision (lines 210-219 in discussion).

Conditional Mfn1 and Mfn2 Knockouts(A and B) Schematic diagram of conditional targeting constructs for Mfn1 (A) and Mfn2 (B) and their homologous recombination into the endogenous loci.  In each case, two loxP recombination sites flank the exon encoding the canonical G-1 GTPase motif. When exposed to Cre recombinase, excision of this critical exon and a frame-shift occur, thus preventing formation of any functional protein. Symbols: black squares, FRT sites; black arrowheads, loxP sites; white arrowheads, PCR primers; DTA, diptheria toxin (PMID: 17693261).

Critique 2: Is anything known about the MFN1 and MFN2 mRNA content in sperm? Can it be excluded that the transcripts play a role during post-fertilization processes? Does the oocyte also contain MFN1 and MFN2 transcripts?

Response: Honda et al. examined the Mfn1 and Mfn2 expression patterns in rat by Northern blot analysis, and found that Mfn1 and Mfn2 were highly expressed in the pubertal and adult rat testis. In addition, they revealed a pronounced expression of MFN2 in round and elongating spermatids by immunohistochemistry (PMID: 14592431). By RT-qPCR, Wang et al. observed Mfn2 mRNA was highly expressed in pachytene spermatocytes and round spermatids, but is nearly imperceptible in Sertoli cells and sperm (PMID: 33674260). Although both Mfn1 and Mfn2 mRNA was highly expressed in oocytes (PMID: 33674260), conditional knockout of Mfn1, but not Mfn2, in growing oocytes results in female infertility (PMID: 30690319).

In our fertility test, adult male Mfn1f/f; Mfn2f/f; Prm1-Cre (Mfn1&2-cKO) were housed with wild-type C57BL/6 females. We have confirmed that Prm1-Cre recombinase efficiency was 100%. Therefore, Mfn1+/-; Mfn2+/-; Prm1-Cre mice should have Mfn1 and Mfn2 deleted in all sperm, but they appeared to be fertile. These data thus don’t support that Mfn transcripts play a role during post-fertilization processes.

Critique 3: Some words are apparently missing, e.g., line 19, line 44.

Response: Thank you for your careful reading. We have revised the text (Lines 19, 45).

Critique 4: Line 64: plays.

Response: Thank you for pointing this out. We have changed “play” to “plays” in revision (Line 65).

Critique 5: Line 73: spermatids.

Response: Thanks. We have changed “spermatid” to “spermatids” in revision (Line 74).

Critique 6: Line 77: It is suggested to replace “revealed” with “suggest”.

Response: Thank you for your suggestion. We replaced “revealed” with “suggest” in revision (Line 78).

Critique 7: Lettering of the organ names in Fig.1c is smaller than in the other parts of the figure.

Response: Thanks. We replaced the previous images with new ones (Line 80).

Critique 8: 2b: Please provide the size of the marker bands.

Response: Thanks. Following your suggestion, we have added the size of the marker bands in Figure 2B (Line 101).

Critique 9: Line 118 ff: TRA-98 and SYCP are nice markers of germ cells and meiotic germ cells, respectively. However, there is no postmeiotic marker shown which would be much more meaningful in the context of this study. Therefore, it is strongly suggested to stain the testes sections with a (late) spermatid marker.

Response: Thanks for your suggestion. We performed IHF with TNP1, which stains the post meiotic elongating and elongated spermatids (Figure S1 and Figure S2 of the revision). IHF revealed that TNP1 signals were readily detectable in testis sections from Mfns-cKO mice, similar to those in control littermates. We have revised the text accordingly in revision (Lines 122-125).

Critique 10: Line 121: We…

Response: Thank you for your careful proofreading. We have corrected this in the revision. (Line 125).

Critique 11: Line 169: … meiosis have been…

Response: Thanks, we have changed “has” to “have” in the revised manuscript (Line 173).

Critique 12: Line 171: I would recommend not to write that a “detailed” analysis has been performed. The analysis is sufficient to say that the proteins are not important for sperm development and for successful fertilization. However, a detailed analysis would require an in-depth-analysis.

Response: Thanks. Revised. (Line 175).

Critique 13: Line 196: …and lead to…

Response: Thanks. We have changed “leads” to “lead” in the revised manuscript (Line 200).

Critique 14: Line 214: “depend”?

Response: Thank you for your careful reading. Revised (Line 225).

Critique 15: Line 216: “om”??

Response: Thanks. “om” is a typo and is deleted from the revision (Line 227).

Critique 16: Page 291: As far as this reviewer knows the Declaration of Helsinki applies only to research with human subjects. Therefore, the Helsinki Declaration may not apply to this mouse study.

Response: Thank you for pointing this out. We revised the Institutional Review Board Statement section (Lines 309-312).

Reviewer 2 Report

Review of Manuscript ID: ijms-1500185, by Miao J. et al., entitled: “MFN1 and MFN2 are dispensable for sperm development and functions in mice that is intended for publication in International Journal of Molecular Sciences

The Authors of the manuscript submitted for review have carried out the study aimed to clarify the role/importance of two mitofusins (MFN1 and MFN2) in sperm development and functions, in the context of their effects on male fertility in mice. Using haploid-specific Prm1-Cre, they showed that mice with single deletion of Mfn1 or Mfn2 in spermatids had normal sperm development and male fertility. Moreover, adult mice with complete deficiency of both mitofusins displayed similar effect. These valuable results were presented in the form of many charts and then reliably interpreted and critically evaluated in the context to data achieved by other investigators. An abundant methodological workshop based on the use of the methods from the fields of molecular biology, experimental embryology/assisted reproductive technology (ICSI) and histology/immunohistofluorescence should also be emphasized. Furthermore, it is noteworthy to mention that the Authors have used the relevant methods for statistical analyzing the results and have selected adequate references.  Generally, the paper is interesting, well written in English, but is not fully prepared in accordance with the requirements of the International Journal of Molecular Sciences.

In my opinion, the following points have to be corrected or added prior to the acceptance of manuscript for publication as has been detailed below:

1) Please add the subsection Experimental design or protocol to the beginning of the Materials and Methods section - is the need for a clear presentation of the different stages of the experiments carried out (in graphical or descriptive form)  

2) Please add the separate Conclusions and Abbreviations sections

3) Please adjust the format of the References section to the requirements of International Journal of Molecular Sciences.

4) An additional item would be completed and linked to section References: 

Samiec, M.; Skrzyszowska, M. Extranuclear Inheritance of Mitochondrial Genome and Epigenetic Reprogrammability of Chromosomal Telomeres in Somatic Cell Cloning of Mammals. Int. J. Mol. Sci. 2021, 22, 3099. doi: 10.3390/ijms22063099

In conclusion, I recommend this manuscript for publication in International Journal of Molecular Sciences, provided that the above-mentioned remarks and suggestions pointed out by the Reviewer will have been taken into consideration by the Authors to the re-edited version of current paper.

Author Response

Response to Reviewer 2 Comments

Comments

The Authors of the manuscript submitted for review have carried out the study aimed to clarify the role/importance of two mitofusins (MFN1 and MFN2) in sperm development and functions, in the context of their effects on male fertility in mice. Using haploid-specific Prm1-Cre, they showed that mice with single deletion of Mfn1 or Mfn2 in spermatids had normal sperm development and male fertility. Moreover, adult mice with complete deficiency of both mitofusins displayed similar effect. These valuable results were presented in the form of many charts and then reliably interpreted and critically evaluated in the context to data achieved by other investigators. An abundant methodological workshop based on the use of the methods from the fields of molecular biology, experimental embryology/assisted reproductive technology (ICSI) and histology/immunohistofluorescence should also be emphasized. Furthermore, it is noteworthy to mention that the Authors have used the relevant methods for statistical analyzing the results and have selected adequate references.  Generally, the paper is interesting, well written in English, but is not fully prepared in accordance with the requirements of the International Journal of Molecular Sciences.

Response: Thank you for your nice comments. Please see our responses to your specific critiques below.

Critique 1: Please add the subsection Experimental design or protocol to the beginning of the Materials and Methods section - is the need for a clear presentation of the different stages of the experiments carried out (in graphical or descriptive form).

Response: Thanks. Following the reviewer’s suggestion, we added a subsection to describe Experimental design at the beginning of the Materials and Methods section (Line 246).

Critique 2: Please add the separate Conclusions and Abbreviations sections.

Response: Thank you for your suggestion. We added the Abbreviations sections in the manuscript (Line 317). According to the Manuscript Preparation Guidelines of IJMS, Conclusions section is not mandatory, and can be added to the manuscript if the discussion is unusually long or complex. In our original submission, we have presented the Conclusions in the Discussion section.

Critique 3: Please adjust the format of the References section to the requirements of International Journal of Molecular Sciences.

Response: The references are prepared and we have checked them for completeness and correctness. The Editorial Office will adjust the References to align with to the format of International Journal of Molecular Sciences.

Critique 4: An additional item would be completed and linked to section References:

Samiec, M.; Skrzyszowska, M. Extranuclear Inheritance of Mitochondrial Genome and Epigenetic Reprogrammability of Chromosomal Telomeres in Somatic Cell Cloning of Mammals. Int. J. Mol. Sci. 2021, 22, 3099. doi: 10.3390/ijms22063099.

Response: Thanks. We cited this paper as Reference [3] in our revised manuscript (Line 325).

Round 2

Reviewer 1 Report

The authors have addressed all my points adequately. Only one recommendation remains:

Line 215: "Deletion of exon 4-6 by ..." should be replaced e.g. with "Deletion of exon 4 of Mfn1 and of exon 6 in Mfn2 by..."